# Rethinking Stormwater: Analysis Using the Hydrosocial Cycle

**Matthew Wilfong ***  **and Mitchell Pavao-Zuckerman**

Department of Environmental Science and Technology, University of Maryland, College Park, MD 20742, USA; mpzucker@umd.edu
* Correspondence: mwilfong@umd.edu

**Abstract:** Water management and governance continues to rely on the scientific and engineering principles of the hydrologic cycle for decision-making on policies and infrastructure choices. This over-reliance on hydrologic-based, technocratic, command-and-control management and governance tends to discount and overlook the political, social, cultural, and economic factors that shape water-society relationships. This paper utilizes an alternative framework, the hydrosocial cycle, to analyze how water and society shape each other over time. In this paper, the hydrosocial framework is applied to stormwater management in the United States. Two hydrosocial case studies centered on rain and stormwater are investigated to highlight how stormwater management can benefit from a hydrosocial approach. The insights and implications from these case studies are then applied to stormwater management by formulating key questions that arise under the hydrosocial framework. These key questions are significant to progressing stormwater management to more sustainable, resilient, and equitable outcomes for environmental and public safety and health. This paper frames a conversation for incorporating the hydrosocial framework into stormwater management and demonstrates the need for an interdisciplinary approach to water management and governance issues.

**Keywords:** stormwater; hydrosocial; interdisciplinary; stormwater management; socio-natural

## 1. Introduction

"The fundamental problem with conventional stormwater management may be the mindset. It does not treat water as a valuable resource but more like a problem to be solved, or even worse, it as a waste product"[1].

Water is a substance that is inextricably linked with life. It is a "non-substitutable flow resource essential for life and ecological health" but also "of deep spiritual and aesthetic significance"[2]. Water and society are deeply connected with water leaving a trace of its historical, political, and social influence on society as it flows over the landscape[3–9]. The "social nature" of water is the idea that water's materiality, conceptual significance, and meaning is the direct result of the social relations that produce it[10]. Social nature reflects Cronon's ideas where, "what we mean when we use the word 'nature' says as much about ourselves as about the things we label with that word"[11]. What people mean when they talk about water, the different names, meanings, and values they place on water are created by socio-natural processes. These socio-natural processes being the internal relations that materially and discursively shape water and society, blurring and abstracting the separation between the two[12].

The idea that water is "inescapably social"[9] is in direct contrast with the preeminent Western epistemology where nature and society are separate entities. This dominant cultural ideology has allowed water to become an object of management, governance, and commodification. As a result,

command-and-control practices and technocratic solutions dominate water management and are the primary mechanisms to control natural hydrologic processes [13–16]. These technocratic solutions struggle to reach resilient, sustainable, and equitable outcomes due to disregarding and overlooking the social nature of water [4–6]. Moving past the command-and-control, engineering-based model is essential to address the complex and wicked problems for a growing population and in an ever-changing climate [13,14,17].

This desocialization and depoliticization of water management under the prevailing Western epistemology is extremely evident in stormwater management in the United States. Stormwater management has been a public health, public safety, and environmental issue throughout the history of mankind, exacerbated by the drastic increase in urbanization within the last century [18]. In the United States, stormwater management remains in a technocratic realm of engineers and hydrologist due to the separation of humans from the hydrologic cycle. The majority of stormwater management and governance decision-making is based solely on hydrologic variables and analyses, rather than utilizing more holistic approaches [15,17,19–21].

Despite this, there has been progress towards more resilient stormwater management—from solely flood control towards treating stormwater prior to release into surrounding waterways. One example of this progress is the utilization of green infrastructure, rather than traditional grey infrastructure, to help manage stormwater volume and quality in urban and suburban areas [13]. Currently though, stormwater management in the United States continues to struggle with changing climatic conditions while maintaining human and environmental well-being [13,14,17]. Many urban water and stormwater management scholars suggest that climate change requires a complete rethinking and overhaul of water management, including stormwater management, especially in urban areas [14,22]. This rethinking of water management parallels the development of the concept of nature-based solutions to urban environmental challenges. Nature-based solutions are ecosystem-based approaches to environmental management that may provide resilient solutions to climate adaptation and mitigation that addresses biophysical, social, and political challenges of implementation and planning [23,24].

Most engineers, hydrologist, and ecologists alike acknowledge that understanding the social, political, and economic factors driving stormwater management is important, but typically, these factors are discounted and not incorporated into decision-making and governance [17,20]. One approach that can help bring these factors into decision-making is the hydrosocial framework, which stresses that water and society exist in an integrated system. So, rather than people affecting hydrologic systems from the outside, the hydrosocial cycle views water and people as an integrated system with internal connections between humans and water [3]. The hydrosocial cycle as a framework for stormwater management can provide the ability to assess and understand the political, economic, social, and cultural dimensions. The hydrosocial cycle promotes a critical analysis of water-society relationships, positioning humans within the hydrologic cycle, where humans and water co-construct themselves based upon complex interactions of social, political, historical, economic, and hydrological factors.

The goal of this article is to explore how the hydrosocial cycle, as a framework for analysis, can provide the platform to investigate the social and natural relations of stormwater. We begin by showing that a hydrological framework that does not integrate the socio-natural aspects of water and stormwater heavily influences most stormwater management thinking and programs. Next, we present details of two case studies to illustrate the application of the hydrosocial cycle through which broader cultural, social, and political factors are linked to water management, in one case rainwater harvesting and the other stormwater management. The insights and lessons learned from these two different case studies are useful and suggestive to what a hydrosocial approach to stormwater management might look like or consider. Finally, we suggest some implications and provide recommendations focused on how hydrosocial analysis of stormwater management can increase the understanding of the socio-natural aspects of stormwater and how stormwater engineers and managers can begin to think within a hydrosocial framework.

## 2. History of Stormwater Management in the United States

Stormwater management is not a modern invention in response to urbanization. Ancient civilizations, like the Romans and Mesopotamians, constructed rather sophisticated water drainage infrastructure throughout their cities [25]. Historically, "pave and pipe approaches" were used to move stormwater off the landscape as quickly as possible with a "slow and soak" approach being utilized currently where stormwater is slowed down and allowed to remain and soak into the landscape over a longer period of time [18]. As populations continue to grow within the United States and throughout the world, a larger proportion of the landscape will be developed into suburban and urban environments. Development of the landscape can have a drastic effect on stormwater hydrology through a host of mechanisms, including removal of vegetation, compaction of soils, and construction of impervious surfaces [17]. The processes of development significantly reduce the ability of the landscape to maintain proper hydrologic functioning [26–31]. As urbanization has continued and the construction of impervious surfaces increased, it has become glaringly evident that stormwater management is necessary to maintain public and environmental health.

The concept of a hydrosocial contract or unwritten contract between society and their government to provide potable drinking water, water sanitation, management of stormwater, and flood protection begins to highlight how society and water have co-evolved over time [32]. This co-evolution can be seen in changes in the hydrosocial contract, especially through the outcomes and goals of stormwater management. Stormwater management in the United States has undergone transitions; however, these transitions has been slow and ineffective at responding to the changing conditions and delivering management outcomes that align with the public and environmental concerns posed by stormwater [13,17,20]. Stormwater management began in urban areas with the primary goal of protecting public health from waterborne diseases that were prevalent due to the dumping of human waste into surrounding waterways. To combat this, some urban areas constructed combined wastewater and stormwater pipes, which transmitted stormwater and wastewater to a central water treatment plant before release into local waterways. These combined sewers work well during dry conditions, but during wet weather, these combined sewers overflow resulting in the direct release of untreated sewage and stormwater directly into surrounding waterways [33]. Additionally, public safety became a primary concern due to flooding resulting from a host of landscape alterations associated with urbanization. To provide flood protection, the dominant view has been to transport stormwater off the landscape as quickly as possible, resulting in the technocratic solution of concrete lining or placing of streams into pipes to expedite the movement of stormwater to larger, receiving bodies of water. This paradigm in stormwater management, characterized primarily by the expedited movement of stormwater off the landscape and into receiving bodies of water, have been called "drained or sewered cities" [13,14].

This paradigm dominated until the beginning of the environmental movement, where society wished to rethink the hydrosocial contract, leading to the subsequent passing of the Clean Water Act (CWA) in the 1970s [17,19]. The passing of the Clean Water Act in 1972 and the subsequent amendments (301 and 402) in 1987 placed legal requirements on state and local governments to control and treat stormwater prior to release into waterways [18]. These policies caused a distinct shift in stormwater management towards stormwater control measures that not only managed the volume, but also the quality of stormwater. Cities and municipalities have aligned with these legal requirements following both traditional grey infrastructure (centralized conveyance systems and water treatment plants) and green infrastructure (GI) (decentralized infiltration systems and practices) with the implementation of either varying spatially or temporally [17–19]. Stormwater management during this era has been called "waterways cities" [13], where the primary goal of stormwater management is to reduce pollutants entering waterways via stormwater through water volume reduction and water quality improvement practices. This is the current paradigm in the United States, but with a vast spectrum of implementation both within cities and throughout the country. Some cities have invested greatly in decentralized GI

while others have continued to rely on grey infrastructure to meet the requirement of the CWA, but the large majority have implemented a complex combination of both grey and GI [34].

While there is agreement that decentralized GI practices will promote more sustainable and resilient stormwater management, the adoption and implementation of these practices has been slow mostly due to social, economic, and political factors [19–21]. Engineers, hydrologists, and ecologists who often make stormwater infrastructure and management decisions frequently overlook these factors [17,20]. This tends to result in the implementation of traditional grey infrastructure rather the adoption of new, GI practices. To compound these issues, climate change has prompted scholars to suggest that a "complete reworking of urban water governance" [35,36] is required to cope with the public health, public safety, and environmental issues. In the United States, stormwater management must adapt to the changing climate, population growth, and increased urbanization towards more resilient, sustainable, and equitable outcomes. This would require stormwater management to incorporate the socio-natural aspects of stormwater into management and governance decision-making. This would also require a conceptual shift away from the hydrologic cycle and towards understanding of stormwater and society more holistically—this transition can be done using the hydrosocial cycle.

## 3. Transition from Hydrologic to Hydrosocial Cycle

Water management and governance has lacked a holistic perspective, when attempting to provide water for societal health, well-being, and prosperity resulting in the tendency to view water as a resource or commodity. This material view of water and water infrastructure has been reinforced by the hydrologic cycle. In the hydrologic cycle, the flow of water throughout the biosphere is a phenomenon pertaining to the "natural circulation of water in, on, and over the Earth's surface" [37]. The hydrologic cycle was first depicted by Robert Horton, an American hydrologist, with the purpose of providing a framework for the continued study of water within the biosphere [37] (Figure 1).

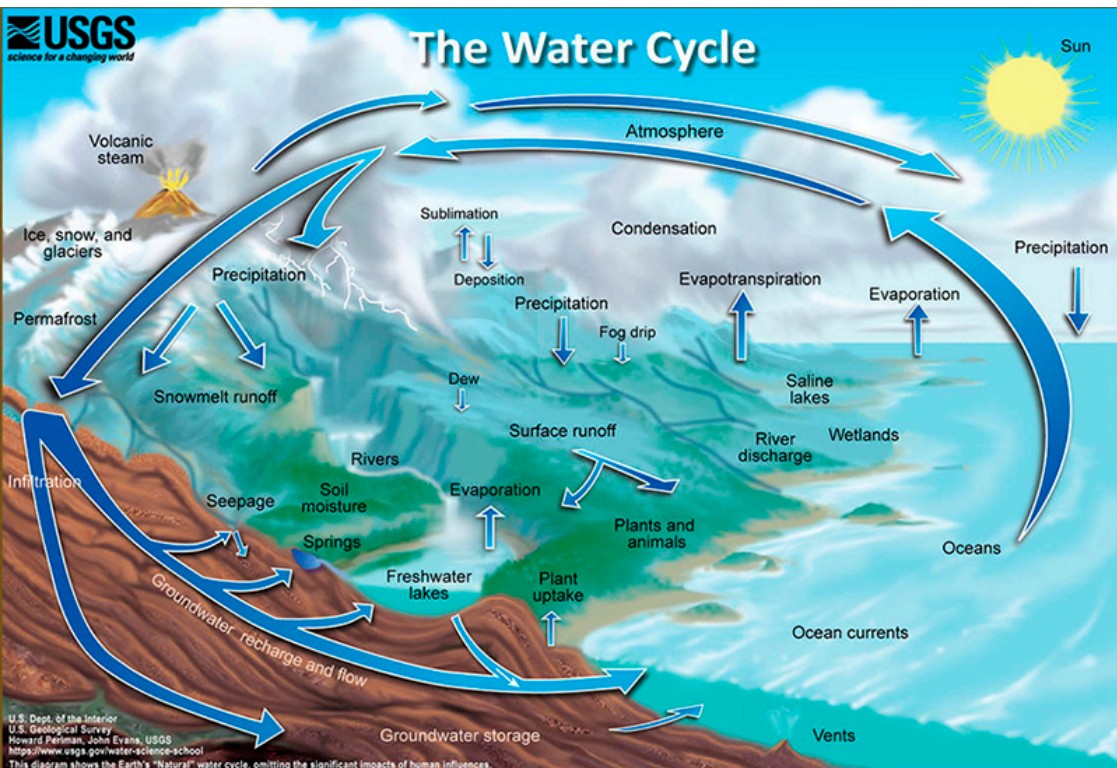

**Figure 1.** An example depiction of the hydrologic cycle seen in most textbooks and taught in introductory environmental classes. This figure demonstrates the separation of humans from the hydrologic cycle. Produced by the United States Geologic Survey.

The hydrological cycle has since become fundamental to understanding the flow of water throughout the biosphere and is still the primary framework presented in introduction hydrology, geology, and environmental science classes [38] (Figure 1). The separation of humans from the hydrologic cycle has persisted despite many scholars identifying the problematic discourse of humans' separation from nature [5,7]. This persistence may be connected to the general lack of representation of humans in hydrologic conceptual models. For example, a recent study reviewed 464 water cycle diagrams from around the world and found only 15% conveyed any type of human interaction within the cycle and only 2% showed any potential effects of human-induced climate change [38]. Excluding human interactions from conceptualizations of water cycles may contribute to mismanagement of water resources and ineffective and inequitable water governance.

The hydrologic cycle leads to the separation of hydrologist from the other stakeholders and variables affecting water management and governance. The reliance on the hydrologic cycle reduces the ability to understand the historical, political, and social dimensions that give meaning, value, context, and power to water in, on, and over Earth's surface. Effectively, the hydrologic cycle "represents water in a way that erases its own social content and operates akin to a mirror of nature, wherein no image of society is reflected back" [9].

The hydrosocial cycle [4] provides an alternative to the widely accepted hydrologic cycle (Figure 2) and broadly conveys how "water" is situated within a continuously adapting cycle shaped by social, physical, and technological drivers. This general framework provides a stark contrast to depictions of the hydrologic cycle. In contrast with the hydrologic cycle, the hydrosocial cycle attempts to understand and account for the historical, political, and social factors that shape water and water management. Rather than separating humans from the flow of water, the hydrosocial cycle captures the reality that "water is simultaneously a physical flow (the circulation of $H_2O$) and a socially and discursively mediated thing implicated in that flow" [4]. The hydrosocial cycle can be defined as "a socio-natural process by which water and society make and remake each other over space and time" [6]. This dialectical relationship between water and society suggests tracing every alteration within the hydrologic cycle to a societal shift of power or structural change is possible [10]. The hydrosocial cycle insinuates that water and society as related internally, each providing meaning, context, and power to the other [6].

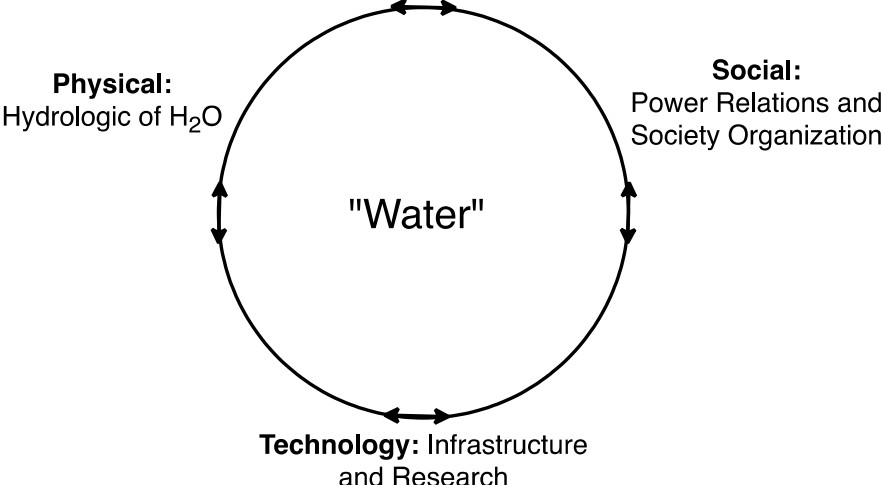

**Figure 2.** A conceptual diagram of the hydrosocial cycle by which the materiality of water, social power and structure, and technology and infrastructure make and re-make "water". Adapted from Linton and Budds, 2014 [4].

The hydrosocial cycle emphasizes the socio-natural aspects of water, particularly where "particular kinds of social relations produced different kinds of water" [5]. These different kinds of "water" arise due to different sociocultural meanings and water-society power relations that produce significant

symbolic and material implications. For example, the sociotechnical processes that create bottled water as an alternative to tap water that citizens would be willing to pay for demonstrates how social, political, economic, and historical factors can create different kinds of water with different values and meanings [7]. In short—the hydrosocial cycle reframes the Western epistemology that divorces nature and society and it allows the analysis of water and society as "the transformations of, and in, the hydrological cycle at local, regional and global levels on the one hand and relations of social, political, economic, and cultural power on the other" [10]. The hydrosocial cycle seeks to understand the socio-natural processes that drive water-society relationships over time and across space.

The hydrosocial cycle can be a powerful framework to analyze the social, political, and historical dimensions of water-society relationships. The key aspects to utilizing the hydrosocial cycle to understand these relations are: (1) water management is necessary to maintain society, and as such, has a substantial driving effect on organizing society and power relations, which then in turn affects the hydrologic flows of water. (2) Water and society are internally related—so that different sociopolitical relations give rise to different kinds of water; and (3) water's material, hydrologic flow, despite being socio-politically altered, still provides important and active processes in the hydrosocial cycle that cannot be discounted [4]. Using the hydrosocial cycle can illuminate previously hidden or obscured nature–water–society relations that when integrated convey how water's production, meaning, value, and context is the product of the coevolution of water and society.

## 4. Methodology: Using the Hydrosocial as Framework for Analysis

The research conducted throughout this article was compromised of a literature review, an in-depth analysis of stormwater-related hydrosocial case studies, and a synthesis of implications that a hydrosocial framework can bring to stormwater management. This research began with a literature review of applications of the hydrosocial framework to analyze water-society relationships in various sociocultural, political, and economic contexts. The literature reviewed spanned a multitude of spatial, cultural, and political settings [6,37–44]. This literature review not only provided a basis for understanding applications of the conceptual framework and theory behind the hydrosocial cycle, but also, a platform to assess and expand into other water-society relationships, like stormwater management.

We formulated a set of questions based upon the literature review that all case studies provided the information to answer. These questions can be used as the basis for assessing the hydrosocial relations for any given water-society relationship. Additionally, they help explore the coevolution of water and society and unveil the internal processes and relationships shaping one another:

1.  What is the definition or conception of water amongst different stakeholders? Do these stakeholders have differing definitions or conceptions?
2.  What is the primary mechanism or driver behind the conception of water for each stakeholder group (i.e., social, historical, economic, political, spiritual, etc.)?
3.  In each instance, how has water and society been co-constructed and internally related to create "different waters in different water-society relationships" [4]?
4.  What are the management and livelihood implications and consequences of the hydrosocial relations between stakeholders and their "waters"?

Each of these questions can be answered differently, depending on the hydrosocial relations present in any location, but they can have a profound effect on the water-society relationship and the overall goals and outcomes of water management and governance. These questions, derived from the literature review, are the foundation of any hydrosocial research and the backbone of beginning to question novel water-society relationships, like stormwater management.

Two case studies were chosen to explore the application of the hydrosocial cycle to analyzing stormwater management in the United States. Each case study analyzed with the above questions demonstrates how water and society continuously make and remake each other through sociocultural

and sociopolitical processes. These two case studies concerning rainwater harvesting in the arid southwest [39] and the political atmosphere surrounding the implementation of green infrastructure (GI) implementation [40] were chosen to be illustrative and representative of the power of the hydrosocial cycle as an analytical framework. Additionally, both case studies were closely related to stormwater and could be utilized to explore what a hydrosocial approach would bring to an analysis of stormwater management. These case studies were utilized as steppingstones to draw equivalents into stormwater management. These parallels allowed for the formulation and articulation of key questions that a hydrosocial framework reveals for stormwater management in the United States.

## 5. Bridging the Hydrosocial into Stormwater Management

### 5.1. Rain Harvesters as "Ethical Desert Dwellers"

In a study of rainwater harvesting programs in Arizona, Lucero Radonic documented how rainwater harvesting produced an intimate connection between residents and rainwater [39]. The city of Tucson instituted a rainwater-harvesting ordinance in 2008 where residents received a $2000 rebate for the installation of cisterns on their property. These rainwater-harvesting practices were readily implemented with nearly two thousand residents installing cisterns in the first six years of the program [39]. The primary goal of the ordinance was to reduce the potable water consumption by residents of Tucson by incentivizing the use of harvested rainwater for irrigation and other household uses. This research revealed that despite the widespread installation of rainwater harvesting practices, potable water consumption by residents did not significantly decline. Radonic analyzed the hydrosocial relations altered and created through the rainwater-harvesting program and determined how and why potable water consumption remained consistent [39].

The goal of decreasing the potable water consumption was not achieved through the rainwater-harvesting ordinance; however, the hydrosocial relationships between the residents, rainwater, and their surrounding environment were transformed. For residents, the rainwater-harvesting program altered their relationship with the surrounding environment, providing a deep connection with the local landscape and water resources. Understanding the hydrosocial relations affords a look into the more nuanced ramifications of the rainwater-harvesting program that go beyond simply lowering potable water consumption.

The rainwater-harvesting program allowed residents to feel as though they were working alongside the natural environment and they *managed their livelihood* within the desert landscape. For example, nearly all residents were utilizing the harvested rainwater to decrease the use of potable water for irrigation purposes. Residents shifted to manually watering the landscape or setting up automated drip irrigation connected to their cisterns, despite being both time and labor intensive compared to typical household watering practices [39]. This alteration in everyday irrigation practices helped produce a tangible connection between the harvested rainwater, the landscape, and the residents, themselves. Additionally, many rainwater-harvesting residents began to replace high water using ornamental plants with more drought-resistance native plants to allow the harvested rainwater to be more efficiently utilized [39]. Residents began to take responsibility for, not only harvesting rainwater, but effectively and efficiently utilizing the rainwater for the betterment of their landscape and to decrease their impact of living within a desert environment. Ultimately, this connection prompted a new socio-natural relationship between the residents and their environment, altering how residents viewed their place on the landscape.

Rainwater is now also *conceptualized as a new "resource"* within the urbanized environment [39]. Many residents, after partaking in the rainwater-harvesting program uprooted their beloved "tropical paradise" landscaping to prevent wasting harvested rainwater [39]. Residents also began citing healthier plants and soil conditions due to rainwater irrigation [39]. Residents started viewing rainwater as higher quality compared to tap water, the result of their intimate relationship with harvesting practices (similar to how a tomato grown in your own garden always tastes better than one

that was store bought). Rainwater became ***socially and culturally constructed*** as a valuable resource that residents could take advantage of and utilize for their own personal benefit and the betterment of the environment as a whole.

The hydrosocial relations that arise because of the water conservation program do not align with the goals and outcomes of the state conversation program, but still provide important, useful insights for future water conservation and management in the face of increased urbanization and climate change in a desert environment. For example, by understanding the hydrosocial relations, one identifies that the usage of harvested rainwater for irrigation is an avenue to promote more efficiency. The vast majority of residents use their harvested rainwater purely for landscape irrigation; however, many residents cited that this practice is inefficient, labor and time-intensive, and particularly wasteful. This wastefulness is due to the inability of residents to closely monitor pumping systems distributing rainwater and/or the forgetfulness of residents to close valves and move hoses and pipes when manually distributing collected rainwater. A hydrological viewpoint may deem the program a failure or advocate for additional rainwater harvesting by residents to lower potable water consumption, but through understanding the hydrosocial relations, the state could help lower potable water consumption through outreach for irrigation practices and promoting the usage of different, more efficient irrigation technologies. This outreach could help residents more efficiently and effectively utilize harvested rainwater and decrease the use of potable water for irrigation purposes when rainwater is either wasted or scarce. Additionally, though, this analysis conveys the socio-natural processes and hydrosocial relationships that arise as a result of the rainwater-harvesting program. These insights are important to understand and assess to determine how residents relate to water resources and their surrounding environment.

*5.2. Co-Option of Green Infrastructure by Grey Epistemologies*

In a second case study, Michael Finewood demonstrates how water management and governance stakeholders have co-opted the conversation around GI for stormwater management to maintain control and power [40]. The city of Pittsburgh, subjected to Consent Decree in 2008, required Allegheny County Sewer Authority (ACSA), in collaboration with municipalities, to improve the quality of water entering the surrounding streams and rivers. The Consent Decree would require large-scale improvements in the stormwater infrastructure within the city costing ACSA approximately $2–4 billion dollars. Stormwater management infrastructure in Pittsburgh would primarily be classified as "grey infrastructure" where combined stormwater and sewage pipes convey water to water treatment facilities prior to release into local waterways. Moving away from grey infrastructure and limiting combined sewer overflows, GI or source-control practices have gained widespread acceptance as a more ecologically friendly and "green" method for stormwater management.

In 2013, ACSA released a "Wet Weather Plan" that detailed how primarily grey infrastructure approaches would be constructed to help meet the Consent Decree that was roundly opposed by a large contingent of the community. Community members instead supported the institution of GI practices across the city to help cope with stormwater during rain events and potentially provide more equitable distribution of benefits from the large-scale infrastructure projects necessary to meet the Consent Decree [40]. The Wet Weather Plan was rejected by the Environmental Protection Agency (EPA), providing community advocates an inroad for the institution of GI practices into the Wet Weather Plan. After the rejection of the grey infrastructure-dominated plan and the backlash faced from the community over the lack of GI, ACSA began to acknowledge the importance of GI to manage stormwater. ACSA understood that the incorporation of GI within the city would be more expensive, require additional planning, and necessitate involvement with the community. To avoid this and maintain the status quo of stormwater management in Pittsburgh, ACSA used their position of power and perceived expertise to control the grey versus GI narrative, and ***how GI was conceptualized***.

Controlling the narrative began when the ASCA began using model analysis to pinpoint "hot spot" areas for GI implementation, allowing ACSA to be viewed as supportive of GI, demonstrating

their expertise for stormwater management, and controlling GI implementation, in general. ASCA also worked alongside community partners to find shovel-ready GI projects. These shovel-ready projects for GI implementation were chosen based on which projects would be most visible to the community, rather than which would provide the highest stormwater management and overall community benefit. ACSA rebranded themselves as "green by mission, green by choice" [40] and began to attend community meetings organized by GI proponents to convey how they were supportive of GI institution. Additionally, ASCA incorporated GI into their revised Wet Weather Plan, but strategically failed to set any specific GI goals or targets. ASCA appeared to the public as honestly incorporating GI into their plan while only superficially endorsing and supporting GI implementation. This could be seen in discussions around construction of GI within low-income neighborhoods where engineers "asked if there was an effective size or type of GI that would not need community feedback" [40].

All of these steps taken by ASCA were to increase their involvement within the GI discussion. Then they began shaping the narrative surrounding GI implementation for stormwater management. ASCA shifted to a more technical narrative for GI implementation centered on hydrology that GI advocates had to embrace to be incorporated within the debate. Community advocates speaking about GI in terms of "water quality compliance" and "long term-maintenance and monitoring," where prior to ASCA involvement, the narrative was centered on broader, less technical ideas, like job creation, economic development, and community improvement. This demonstrates co-opting of the GI narrative by ASCA. ASCA utilized their position of power and control to infiltrate the GI discussion, shift the narrative to benefit their viewpoint, and ultimately, converge the narrative more towards their preferred grey infrastructure, technological-dominated views. By doing so, ASCA could not only control what infrastructure is built (grey vs. green), but where it is built and who it benefits. Here, ***stormwater and society co-construct each other*** to reconceptualize GI through ***political, economic, and socially*** distinct narratives.

ASCA was not only shaping the narrative surrounding stormwater management, but also around larger, broader urban governance issues. The converging of ASCA's engineering-based ecohydrologic narrative with community-based involvement in stormwater management through GI did not result in collaboration and erosion of epistemological difference, it simply reframed the city as slightly greener, but the dominant existing command-and-control, technocratic regime remained in control and power [40]. This results in the same stakeholders "shaping, controlling, and reproducing the city" [41] to align with their interests, benefiting themselves, and neglecting other, but behind the shroud of collaboration with community groups. Community members wanted GI to be incorporated into the Wet Weather Plan to provide multi-functional benefits, especially for low-income neighborhoods, who disproportionally are affected by the multitude of environmental harms of urban living [40]. This shift would entail a reworking of urban water management within Pittsburgh and potentially a removal of powerful actors, like ASCA. To prevent this, ACSA co-opted the messages and views of GI proponents and community members, using their position of power and expertise to control the narrative. This allowed ASCA to maintain the status quo for urban water management, and more largely, produce a city designed to advantage certain actors and stakeholders, and neglect others, which has distinct ***management and livelihood implications***.

A hydrosocial lens allows an analysis that better understands the sociopolitical drivers that maintained a grey epistemology within Pittsburgh water governance and management. The dominant ecohydrologic view allows for the perpetuation of dominant management practices and outcomes and the co-option of the GI narrative. Through an analysis of the hydrosocial relations, it becomes evident that GI advocates and community members must change how they approach GI implementation and adoption to achieve their goals. GI advocates must begin to ask deeper, more complex questions of urban power dynamics, human-nature integration, and capitalistic endeavors, and if progress towards GI adoption wishes to overcome the dominant, deeply engrained technocratic management, especially surrounding urban water management.

*5.3. Synthesizing Case Studies*

These case studies demonstrate the significance of understanding the hydrosocial relations concerning stormwater management and governance. When used as an analytical tool, approach, or lens, the hydrosocial cycle illuminates previously obscured or invisible social, political, historical, and/or economic interactions that shape how the framing of stormwater and how governance and management is undertaken and supported. The findings and implications of each case study and how each case study answers the guided hydrosocial questions mentioned previously is depicted in Table 1. We summarize the insights from the hydrosocial analysis, the implications of these insights, and how these insights are different from typical hydrologic or ecohydrologic research in the following section (see Table 1).

**Table 1.** Comparative Insights from Applying Hydrosocial Framework. Green Infrastructure (GI) and Allegheny County Sewer Authority (ACSA).

| Insights | Tucson, Arizona | Pittsburgh, Pennsylvania |
|---|---|---|
| Definition or Conception of Water | Rainwater—Water as "resource" | Stormwater—Water as "hazard" |
| Primary Mechanism or Drivers | Social, Physical, Political | Political, Social, Physical, Economic |
| Co-construction of Water and Society | State rainwater harvesting program designed to promote decrease in potable water consumption by residents. Program alters residents' connection with rainwater, tap water, and surrounding landscape. | City needed to adhere to Consent Decree to manage stormwater within Pittsburgh. Conflict between GI proponents and ASCA on how to best address stormwater and adhere to Consent Decree. |
| Insights of Traditional Ecohydrological Analysis | Rainwater harvesting program failing to decrease potable water consumption among residents. Increased adoption could decrease potable water usage. | Grey infrastructure is most cost-efficient choice to manage stormwater in Pittsburgh, but GI implementation in "hot spot" areas for ecological benefit. |
| Insights Using Hydrosocial Analysis | Rainwater harvesting program failing to decrease potable water consumption among residents. Residents as "ethical desert dwellers," not as economic rational decision-makers that use rainwater harvesting to validate their decision to live in desert environment. | The co-option of GI for stormwater management by those in power to maintain authority over water management. GI narrative controlled by traditional technocratic, command-and-control water management regime. |
| Management and Livelihood Implications | The implementation of future conservation programs towards more efficient, effective usage of collected rainwater and other programs to decrease public potable water usage. | GI advocates should acknowledge how they have lost control of GI narrative. Begin to ask larger questions of urban power dynamics to unseat traditional institutions, power dynamics, and epistemologies. |

In Tucson, Arizona, state-sponsored rainwater harvesting programs fostered an intimate connection for residents with their desert landscape; however, it did not produce the desired outcome of lowering potable water consumption. At first glance, the rainwater-harvesting program was a failure; however, by understanding the changing hydrosocial relations, future management decisions and programs can be more successful. In Pittsburgh, Pennsylvania, the co-option of GI by engineers and hydrologist to maintain the status quo and perpetuate environmental inequality within the city's stormwater management plan is significant to acknowledge. The hydrosocial configurations, despite appearing to favor a shift to supporting GI and the management of stormwater as a "resource" were purely superficial. To truly progress towards the implementation of GI, co-option must be understood through hydrosocial relations, and avoided, and overcome. These case studies provide

valuable information on how the hydrosocial framework illuminated the often hidden social, cultural, and political factors underlying water-society relationships. Importantly too, these case studies provide direct relations to implementing a hydrosocial framework into stormwater management in the United States and what questions would arise from doing so and the implications of those questions.

## 6. Discussion: Applying the Hydrosocial Cycle to Stormwater Management

Stormwater management in the United States is distinctly a socio-eco-technical issue [13,14,21]; however, the current solely hydrology-focused paradigm depoliticizes the management and governance of stormwater. Within the past decade, there has been substantial pressure from environmental and social advocates to transition stormwater management towards more sustainable, resilient, and equitable goals and outcomes [16,17]. Despite this pressure, the paradigm shift and evolution has been markedly slow and, in some cases, non-existent [14,42,43]. There is substantial knowledge that social, political, economic, and historical factors underpin and affect stormwater management; however, they are rarely incorporated into management and governance, maintaining the status quo of stormwater management being an apolitical, asocial, and ahistorical process [13,14,17,20,25,42–44].

These cases studies provide interesting and useful parallel insights for the application of into the hydrosocial cycle framework to stormwater management. For example, in Arizona, the way in which the residents related, viewed, and conceptualized rainwater and their place on the landscape shifted due to rainwater harvesting. Socially and culturally, rainwater began to be seen as a significant resource in the desert landscape that should be efficiently harvested and utilized by the residents living there. Similarly, how stormwater is socially and culturally constructed highly dictates how it is managed across the United States. By understanding the social and cultural factors influencing how stormwater is viewed and conceptualized, infrastructure and management plans can be tailored to help shift the conceptualization or simply work within the bounds of a given conceptualization [26]. In Tucson, by providing a tangible, intimate relationship between the residents and rainwater, rainwater was elevated into a socially and culturally important resource. Perhaps doing the same for stormwater will help shift the narrative away from "stormwater as a pollutant" and towards "stormwater as a resource." Undoubtedly, understanding the social and cultural factors influencing stormwater are important and answering the question of "what is stormwater?" both socially and culturally is paramount to successful transitions in management paradigms.

In Pittsburgh, the ability of powerful stakeholders to control the narrative around infrastructure choices, particularly for stormwater management, demonstrated the importance of political and economic drivers. ASCA could utilize their seat of power and influence to dictate what infrastructure was built, where it was built, and who is was built to benefit. The power and influence of ASCA arises due to their positioning within the hydrosocial relations and ability to control the narrative to frame their positions and discount others. Only through an understanding of these hydrosocial relations could the discursive framing employed by ASCA be assessed and potentially overcame. This case study draws strong connections with stormwater management in the United States, especially the political and economic aspects that are often overlooked. In Pittsburgh, the stormwater management infrastructure choices were controlled through co-option of narratives to keep certain stakeholders in power and remake the city in the image of their desires. Similarly, stormwater management across the United States provides a platform for actors to control what infrastructure is built for stormwater management, where they are built, who reaps the benefits of the infrastructure, and who is neglected. Stormwater management in the United States is highly political and investigating the hydrosocial relations, answering questions like, "where are stormwater management practices built and why?" will begin to promote a transition away from the technocratic management paradigm dominating stormwater management throughout the United States.

Stormwater management provides an excellent platform for the application of the hydrosocial cycle framework of analysis to better understand why the paradigm shift in management has faltered and identify opportunities for progressing stormwater management to the more desired state towards

sustainability, resilience, and equity. Engaging with the hydrosocial framework for stormwater management raises some important questions that will undoubtedly shape the ecological, social, political, and economic outcomes for the future of stormwater management in the United States. These questions are related to the framing questions we identified from the literature review (above) and include:

1. ***Conception and Definition of Stormwater—what is stormwater? Is it a natural resource or a pollutant?***—Stormwater tends to be seen as nuisance, hazard, or "a problem to be solved" [1] rather than a natural resource that can be a "remedy to ongoing water resource challenges and constraints" [35]. How will climate change (increase in droughts, flash flooding, etc.) affect this? Can stormwater be "re-made" as a natural resource, and if so, will this reconceptualization begin a new paradigm in stormwater management [26]?

2. ***Co-construction of Stormwater and Society—how does the legal structure frame stormwater?***—The Clean Water Act has been described as a "liability, not a tool to manage stormwater—giving cities the responsibility, but not the authority to control stormwater from private property" [17]. Can changing the CWA or introducing new legislation shift the discursive framing of the management of stormwater from a "liability" to an "opportunity" for cities and communities?

3. ***Co-construction of Stormwater and Society—where are stormwater management practices built and why?***—Stormwater does not occur uniformly across the landscape, and it "is rarely a medium of rigid social structures" [4]. As a result, there is a disconnection between political and hydrologic boundaries for management, including public versus private land. Is integration of political and hydrologic units possible? Will shifting the responsibility to "private landowners who generate stormwater by changing their land features," [17] provide more or less integration?

4. ***Management and Livelihood Implications—who benefits from stormwater management?***—Centralized and decentralized GI stormwater best management practices as more sustainable, equitable solutions for stormwater management have garnered considerable attention. Are governments utilizing GI, along with neoliberal ideology, to maintain power and authority over the landscape of urban areas? Can we provide truly sustainable and equitable solutions for stormwater management?

Each of the questions above arise through an analysis of the hydrosocial relations of stormwater management in the United States. These questions are primarily social, political, or economic, obstacles or impediments rather than scientific knowledge gaps or concerning the physical nature of water (hydrology). Many ecologists understand that "we already have many of the technologies to address the problem of stormwater runoff" [20]; however, it is an insufficient understanding and accounting for the socio-natural processes of stormwater that hinder progress. The hydrosocial cycle as an analytical framework provides the foundation to begin to answer these questions. Each of these questions requires an in-depth analysis of the internally related processes between stormwater and society that have shaped stormwater, socially and discursively. The goal of employing the hydrosocial as a framework for analysis is to understand how stormwater socially, politically, and historically and the implication this can have on future management decisions. By understanding the different stormwater-society relationships that give rise to the different definitions and conceptions of stormwater, a better understanding of the obstacles and identification of potential avenues to alter these relations is possible.

For stormwater managers, these questions prompt a re-thinking of the management of stormwater. If stormwater is a resource, how can stormwater managers provide infrastructure to best utilize the potential of stormwater for homeowners, industry, government, and businesses alike? If the goal of stormwater management is resilience and equity, how can stormwater managers incorporate environmental and social equity into their management plans to be sure that decisions and implementation of stormwater management practices are equitable? An adoption of the hydrosocial

framework will create stormwater managers who think more critically, holistically, and collaboratively with the communities. Stormwater management in the United States is strongly dictated by power and authority, potentially through this framework, inequalities and injustice that tend to dominate environmental management can be identified and avoided for stormwater management.

For hydrosocial researchers, stormwater can provide a new avenue to understand more complex nature-society relationships. For example, stormwater is difficult to manage, as with other environmental issues, due to the disperse nature of stormwater across the landscape and the difficulty in managing private versus public lands. By understanding the hydrosocial dynamics that prevent the management of stormwater on private lands, hydrosocial researchers can begin to, more broadly, investigate the nature-society relations that arise due to private land. Additionally, stormwater can be a means for hydrosocial researchers to investigate how the framing of water in different contexts, alters how it is managed (i.e., stormwater/rainwater as a hazard, pollutant, nuisance versus as a natural resource). A hydrosocial framework for stormwater management has significant implications for both stormwater managers and hydrosocial researchers and provides a platform for collaboration between the two.

## 7. Conclusions

Many ecologists and engineers suggest that the technology to achieve more sustainable, resilient, and equitable water management in cities is available through stormwater GI, low-impact development, and best management practices [17,20]. However, they understand that implementing these technologies is relatively futile without social, political, and economic acceptance and support [34,42,45,46]. Hydrosocial research provides the foundation to increase the successful implementation of stormwater management technologies and practices within a diverse range of hydrosocial configurations. Natural and social scientists alike can utilize the hydrosocial cycle, bringing stormwater management out from behind the technocratic shroud of the hydrologic cycle and past the nature-society dualistic relationship.

Stormwater provides the basis to understand, more broadly, urban life and inequity through a hydrosocial framework. For hydrosocial researchers, stormwater is a medium within water–society relationships that has immense research potential, specifically for improving the resiliency, sustainability, and equity of stormwater management in the United States. Stormwater provides an excellent platform to see how application of the insights and implications from research can be utilized for meaningful and relevant changes to stormwater management outcomes [47]. Stormwater managers often encounter political, social, and economic obstacles, which are difficult to address when attempting to provide the best stormwater management outcomes for public and environmental health. The hydrosocial cycle provides the foundation to place ecohydrologic research into the a more holistic setting, promoting reflexivity in research, framing advances in technologies or management within the appropriate and necessary social, political, and economic climates.

Stormwater and the management of stormwater is highly cultural, social, and political in nature and only through incorporation of these factors into management decision-making and governance, can a transition towards more sustainable, resilient, and equitable stormwater management be reached. It is suggested here that in the short-term, hydrosocial analyses on stormwater management will be necessary in promoting a more resilient, sustainable, and equitable stormwater management paradigm. Ultimately, the hope is that stormwater will become an environmental flow within the hydrosocial cycle assessed, understood, and managed by engineers, ecologists, hydrologists, political ecologists, economists, and geographers alike.

**Author Contributions:** Conceptualization, M.W.; Methodology, M.W.; Investigation, M.W.; Writing—Original draft preparation, M.W.; Review and Editing, M.P.-Z. All authors have read and agreed to the published version of the manuscript.

**Funding:** This research was supported in part by NSF-CNH/AGS no. 1518376 and a USDA NIFA Hatch project through the Maryland Agricultural Experimentation Station.

**Acknowledgments:** We are extremely grateful for the guidance and mentorship provided by Michael Paolisso throughout the conceptualization, investigation, writing, and editing of this paper. Additionally, we would like to thank Sarah Ponte Cabral, Amanda Rockler, and Marissa Mastler for their edits and reviews throughout the writing process. Finally, we would like to thank the anonymous reviewers for improving the quality and extending the scope of this manuscript.

**Conflicts of Interest:** The authors declare no conflict of interest.

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
