# Peer review of "Rethinking Stormwater: Analysis Using the Hydrosocial Cycle"

_water, doi:10.3390/w12051273_

Round 1

Reviewer 1 Report

Dear Authors 

Thank you for a well-written and important article overall in the field of water management and governance.

My main points for you to consider are as follows:

  1. Proof-read the article one more time to correct writing and grammatical errors in some places e.g. on Page 2 last paragraph you use 'we' and 'I' interchangeably; on page 3 the penultimate paragraph 'has' should be 'have' ; on Page 10 last paragraph 'overcame' should read 'overcome' etc
  2. Section 4 needs to make clearer that the paper is based on a a literature review and desktop case studies. Perhaps also mentioning that this section is the Methods section might also help the reader.
  3. Conceptually it might give the paper a wider audience if they linked their paper to the wider Nature-Based Solutions conversations currently going on in European cities. You could also consider making some use of the 'hydrosocial contract' concept by Turton & Meissner to dovetail your argumentation for water management that takes fully into account the relations between nature and society and between different stakeholders within societies as well. 

Reviewer 2 Report

The presented paper is related with an interesting issue, however it has some aspects that have to be improved and changed before publication. The paper extension is big, too much importance (and text) is dedicated to revision, and these should be reduced. The case studies should occupy the must part of the paper. For example the history should be a paragraph and not two pages...The reviewer does not understand the need of divide the bibliographic review in so many items...maybe reducing each one of them it would be fit in the introduction of the paper. 

Avoid transcriptions in this type of publications. The authors may reference but write the content in their own words.

Figure 1 is unnecessary and somewhat ridiculous in a research paper.

Never use the first person (I- page 2, or we all over the text).

Comment: Harvard system for references is the most used for a reason.

Reviewer 3 Report

Generally: The topic of this paper is relevant, timely, interdisciplinary and of interest to the audience of this journal. The paper is based on academic standards. The content of this paper is technically accurate and sound. The abstract is concise and sufficient. The introduction provides the necessary background information. The research methodology for the study is appropriate and applied properly. The results of analysis are correctly interpreted and conclusions are sound. References are complete and appropriate. 

Comments: Tables are not used. Only very few of figures. Two times is used Fig. 1

There are missing some numbers of results in the described case studies - reference is not in a very known journal in the field of Water sciences

Round 2

Reviewer 2 Report

The reviewer accept the paper in the present form after minor spelling revisions.